# An Improved Performance Radar Sensor for K-Band Automotive Radars

**DOI:** 10.3390/s23167070

**Published:** 2023-08-10

**Authors:** Anwer S. Abd El-Hameed, Eman G. Ouf, Ayman Elboushi, Asmaa G. Seliem, Yuta Izumi

**Affiliations:** 1Electronics Research Institute, Cairo 4473221, Egypt; emanouf@eri.sci.eg (E.G.O.); a_m_fekry@eri.sci.eg (A.E.); 2Communication and Computer Department, Faculty of Engineering, Nahda University, Beni Suef 62746, Egypt; asmaa.seliem@nub.edu.eg; 3Muroran Institute of Technology, Hankkaido 0500071, Japan; yizumi@mmm.muroran-it.ac.jp

**Keywords:** automotive radar, MIMO, PGAA, SRR, superstrate, UWB

## Abstract

This paper presents a new radar sensor configuration of a planar grid antenna array (PGAA) for automotive ultra-wideband (UWB) radar applications. For system realisation, the MIMO concept is adopted. The proposed antenna is designed to operate over the 24 GHz frequency band. It is based on split-ring resonator (SRR) elements to enhance the operating bandwidth and increase the antenna gain, leading to a better-performing radar system. The PGAA consists of thirty-one radiating elements, in which each element excitation is obtained using a common transmission line centre fed by a 50 Ω coaxial probe. By introducing a superstrate dielectric layer at a distance of λ/2 from the top of the antenna array, the PGAA gain and impedance bandwidth are further improved. The gain is improved by 2.7 dB to reach 16.5 dBi at 24 GHz, and the impedance bandwidth is enhanced to 9.3 GHz (37.7%). The measured impedance bandwidth of the proposed antenna array ranges from 20 GHz to 29.3 GHz for a reflection coefficient ($S_{11}$) of less than −10 dB. The proposed antenna is validated for automotive applications.

## 1. Introduction

The automotive radar is one of the numerous sensor systems used to avoid collisions, with cars traveling at high speeds in the opposite direction. The radar initially detects a fast-moving object and determines its distance. If the distance is crucial, i.e., a collision cannot be averted, the radar sends a signal to activate the protective mechanism. Automotive radar antennas should have a low beamwidth of around 15° in the elevation plane and a broader beamwidth of about 60° in the azimuth plane [1,2,3,4]. Furthermore, the waves with orthogonal polarisation from automobiles traveling in the other direction do not affect radar performance.

One of the most important components of any wireless system is the microstrip antenna [5,6]. For use in automotive radar applications, Kraus suggested the GAA in 1964 [7]. Conti et al. deployed a grid antenna array (GAA) as a standing wave antenna in microstrip technology in 1981 [8], while Zhang and Sun resurrected the GAA in the development of antenna-in-package technology in 2009 [9]. Chen et al. used a parallel evolutionary algorithm to optimise the grid array antenna in 2010 [10]. They demonstrated that the GAA in microstrip technology has all of the features of traditional patch-type radiators, plus large bandwidth, high gain, and appropriate cross-polarisation control [6,8]. Because of its features in terms of gain, bandwidth, feeding simplicity, and construction [10,11], the GAA is a better candidate for 60 GHz radio than the conventional patch radiators. Generally, the GAA is designed for fan- or pencil-beam radiation [7,12].

A few academics have sought to improve the GAA bandwidth performance. These antennas, however, have been developed and proven at C-band frequencies. The researchers in [13] utilised elliptical radiating components to enhance the impedance bandwidth of the GAA. The same researchers exhibited further bandwidth gain in the proposed GAA in another study [14], where traditional straight-line transmission lines were substituted by sinusoidal lines. The GAA with sinusoidal transmission and elliptical radiating lines gave an impedance bandwidth of 627 MHz and radiation bandwidth of 400 MHz, respectively. To the best of the authors’ knowledge, GAA bandwidth enhancement at the 24 GHz frequency spectrum has only been reported in [10,11,12,13,14,15,16]. However, antenna gain remains an issue. For gain improvement, one grid and various open-ended stubs were formed [17], which had a gain of 11.5 dBi and a 10 dB return loss relative bandwidth of 3.66%. However, the unit cell size was larger than the free space wavelength. In order to make the GAA smaller, a bent grid antenna was proposed in [18]. The long lines were folded, in contrast to a traditional grid antenna, and overall, one unit cell was smaller than a free space wavelength. In addition, the long bent lines contributed to improving the radiation. At 35 GHz, this antenna had a high gain of 9.4 dBi. A high-gain antenna system is needed to counteract the signal attenuation that occurs at millimetre-wave frequencies owing to oxygen molecule absorption in addition to the aforementioned factors. The use of an antenna array with a suitable feeding network is one of the primary gain-enhancing strategies. Superstrate technology can also be used for further gain augmentation [18,19]. Therefore, wide bandwidth, small size, low profile, flexibility, and high gain are necessary.

Recently, the MIMO technique has been of great interest due to its impact on the design of high-angular-resolution radars. A high resolution is needed for the realisation of a virtual array that contains a large number of antenna elements. The number of virtual antenna elements is the outcome of multiplying the number of receiving antennas by the number of transmitting antennas. Consequently, a larger virtual aperture is produced compared with a physical antenna [20,21]. Based on the maximum range obtained with the MIMO radar, three different categories are possible for the MIMO radar: long range, medium range, and short range. In [22], 2Tx-4Rx 1D and 3Tx-4Rx 2D MIMO radar patch antenna arrays were proposed. A long-range object detection radar with virtual array distribution was reported for a high-resolution multi-input–multi-output (MIMO) system [23]. Another MIMO antenna arrangement based on a sparse array that can simultaneously cover a wide field-of-view (FOV) and achieve the required azimuth resolution at different ranges is explained in detail in [24]. The discussed configuration can cover both the near and far range. Although all the abovementioned papers describe different MIMO configurations, they either ignore the antenna design and only consider ideal elements or suffer from low antenna gain.

This paper introduces a wideband microstrip GAA excited using a coaxial-fed probe with a superstrate dielectric layer operating at 24 GHz for automotive UWB radar applications. A superstrate is added above the radiating elements to increase the antenna gain by 2.7 dB to reach 16.5 dBi. More importantly, the side lobe level (SLL) is reduced to less than −15 dB. The suggested design has a simple feed, a broadside radiation pattern, and few optimisation parameters. In both the measured and calculated results, the suggested PGAA can achieve a gain of 16.5 dBi and a reflection coefficient |S11| of less than −10 dB throughout the frequency range of 20–29.3 GHz (about 37.7% bandwidth). The performance of the suggested design fulfils the criteria for an automotive radar, which is verified via system-level simulation.

## 2. MIMO Concept of Radar

Recently, the concept of MIMO has been presented for ground-based synthetic aperture radars (GB-SARs) and automotive radars [5,20,25,26]. The idea of MIMO radar systems, which is presented in Section 5, is based on transmitting multiple signals from multiple Tx antennas which are received by all Rx antennas to synthesise a relatively large antenna aperture. All the received signals are used to reconstruct SAR images based on different kinds of algorithms. The back-projection algorithm is the most common algorithm [5,27,28]. To construct an antenna array with a low side lobe level, the distance between the array elements should be less than half the wavelength, which is practically impossible in most cases. Employing the MIMO approach is the best way to tackle this problem because the signal transmitted from one antenna and received by another is proportional to the multiplication of their radiation patterns, i.e., the virtual array concept. As a result, if the MIMO configuration is carefully arranged so that each Tx/Rx radiation pattern null faces Rx/Tx peak, the SLL will be drastically reduced, as we will discuss later in detail. For a better understanding, assume n and m transmit and receive antennas are, respectively, are used to construct a MIMO radar system. The convolution of the corresponding excitation distributions results from the superposition of the observations, as shown in Figure 1. This arrangement produces a synthetic antenna array with densely spaced-out subarrays. Accordingly, the virtual array SLL is significantly reduced, providing immunity to unwanted clutter. Other transmit and receive antenna configurations produce an even wider aperture. This paper presents a simple alternative MIMO antenna configuration based on the GAA geometry to replace the commercial Ka-band conventional patch antenna. The proposed antenna has the advantages of stationarity, good radiation performance, light weight, and low cost, and follows the automotive radar antenna radiation pattern standards mentioned in the Introduction.

## 3. Antenna Design

Figure 2 shows the proposed PGAA printed on a RO3003 dielectric substrate with a relative permittivity of 3 and a thickness of 1.527 mm. Conventional radiating lines have poor narrowband performance [29]. As a result, conventional short radiating lines are tuned to provide broadband performance. The SRR radiating elements are chosen for their popular features of wide impedance bandwidth and high gain [30]. The SRR consists of two conductive rings separated by a tiny gap that is printed on the same dielectric substrate. Each ring contains a slit, and the rings are arranged so that the slits are on opposite sides of the line of symmetry. In this sense, the SRR can be thought of as two interconnected tiny loop antennas operating in two adjacent frequency bands. As a result, the antenna can provide a greater bandwidth. The optimised dimensions of the proposed antenna are given in Table 1.

The SRRs are designed at the frequency of interest so that the current directions on the array are generally as shown in Figure 3. Each SRR current is in phase, and each horizontal transmission line can carry a full wavelength current element. Therefore, the phase of the vertically polarised field components can be changed to augment, while the phase of the horizontally polarised field components cancels. The grid structure can be used as a travelling wave device, as described by Kraus and Tiuri et al. [7], or as a resonant (broadside) radiating structure with the appropriate frequency, feed point, and SRR parameters. This study focuses on the latter use of a resonant radiating structure.

### 3.1. Design of SRR Element

The basic principles of the proposed design are clarified through a detailed analysis of a single SRR and the grid structures. The SRR consists of two conductive rings that are printed on a Rogers substrate and positioned with a small gap between them. The two rings in the SRR have slits, and they are arranged so that the slits are positioned on opposite sides of the line of symmetry. The dimensions of the SRR are considerably smaller than the wavelength. Taking this into account, the SRR can be conceptualised as two small loop antennas coupled to each other.

The excited voltages are treated as two discrete voltage sources at points A and B, denoted as source Vo and source Vi, respectively. Figure 4a depicts the analogous circuit in its entirety. The symbol $C_{slit}$ represents the capacitance of the slots, where R′ and L′ represent the distributed resistance and inductance, respectively. C′ refers to the distributed capacitance of the gap between the rings. Two primary loops—the inner and the outer rings of the circuit shown in Figure 4a are connected across the gap by the distributed capacitance C′.

The ratio of the areas created by the outer and inner rings, respectively, shows that voltage Vo is always higher than voltage Vi. As a result, the current virtually crosses the gap from the outer ring to the inner ring via several branches created by the dispersed capacitance C′. This branching causes the current in the outer ring to vary according to its location. It reaches its maximum value at point A and then starts to decrease along the ring before becoming negligible at the gab. Of course, any current that flows from the outer ring into the inner ring contributes to the total inner-ring current. As a result, the current in the inner ring is greatest at the position of the voltage source Vi (point B), and then it gradually decreases along the ring until it reaches its lowest value at the slit. Since the net-ring current is barely affected by the current flowing through the slit ($C_{slit}$), $C_{slit}$ can actually be considered to be zero. Since the current flows primarily through the slit, the entire circuit can be approximated using the much simpler circuit shown in Figure 4b. This consists of a single voltage source and a serial tank circuit. In fact, a single, electrically tiny loop antenna loaded with a capacitor should perform very similarly to the SRR.

The radiator arrangement is the only factor that controls the structure’s radiation characteristics of the structure. The SRR resonance frequency is considered at 24 GHz. The parameters of the basic structure are extracted from the simplified equivalent circuit shown in Figure 4b. The values of the inductance L and C are extracted according to Equation (1).
(1)$$f_{o}=\frac{1}{2\pi \sqrt{LC}}$$

The value of L is supposed to be 0.875 nH, so the value of C is calculated as 0.05 PF. To obtain a low-quality factor, the value of R is 10 ohms. Accordingly, a single element of a planner SRR antenna was simulated using ADS software, as shown in Figure 4. Figure 5 compares the reflection coefficients of the CST design and the RLC circuit model, which were extracted using ADS software, and the results show good agreement. The reason for neglecting the effect of the upper termination line shown in Figure 2 in the single-element equivalent circuit is due to the presence of an opposing current in the corresponding lower section. This opposing current leads to a cancellation effect, as shown in the electric field distribution in Figure 3, resulting in the negligible contribution of the upper termination section to the overall circuit behaviour. Figure 3a presents the electric field distribution of a single element, emphasising that even when simulating a single element, the influence of the horizontal termination section is minimal and can be ignored.

### 3.2. Grid Antenna Array Results

Two simulators were used to verify the simulated prototype results: CST Microwave Studio Version 2021 and HFSS Version 2021. The voltage standing wave ratio (VSWR) of the proposed GAA is demonstrated in Figure 6, which shows a wide bandwidth ranging from 20 GHz to 26 GHz with a magnitude of <2 dB. A very good agreement between CST and HFSS results is observed.

Furthermore, the realised gain radiation patterns are depicted in both the E plane and the H plane in Figure 7. The proposed GAA antenna shows a half-power beamwidth (HPWB) of 18 ° in the E plane and 69 ° in the H plane at 24 GHz. The gain over the frequency shows an average value of 12 dBi, as shown in Figure 8. It is seen that the gain at 24 GHz is almost 13.8 dBi.

### 3.3. Bandwidth and Gain Enhancement Using a Superstrate Dielectric Layer

In general, automotive radars have to operate in temperatures ranging from 40 °C to 85 °C or even higher. They must tolerate shock and vibration and sometimes operate in wet, icy, or other adverse weather conditions. Therefore, the choice of materials, manufacturing techniques, packaging, and sensor installation is important. Materials used as substrates must be suitable for sub-millimetre-wave frequencies and have physical and electrical properties that remain mostly unchanged over a wide temperature range, and they should not absorb moisture. On the other hand, standard microwave substrates can be somewhat expensive, so a trade-off must be considered. Standard printed circuit board (PCB) processes should be used to etch the planar structures as far as possible. However, the 24 GHz frequency range may require accuracies down to 200 µm, which also presents a significant problem for optimum antenna design. In addition, the right coating must be used to prevent corrosion on metal surfaces. A Rogers RO3003 material with a relative permittivity of 3 and a thickness of 1.527 mm was therefore chosen for both packaging and gain enhancement. The dielectric layer is located at a distance of approximately (λ)⁄2, corresponding to 6 mm from the top of the antenna array substrate, as shown in Figure 9. Numerous reflections between the radiating element and the dielectric slab result in gain enhancement and simultaneous projections of the radiators. In other words, the radiation intensity of the antenna's primary beam is focused using supertasters as a lens, resulting in an apparent increase in antenna gain. The presence of superstrates does not affect the area of the automotive radar as there is sufficient space in the z-direction. The effect of the dielectric superstrate on the antenna bandwidth is shown in Figure 9. The impedance matching is improved and covers the frequency range from 20 GHz to 29.3 GHz with a VSWR level of <2. Figure 10 illustrates the normalised realised gain radiation patterns, with an HPBW of 15 deg in the E plane and 60 deg in the H plane, and an SLL of −15 dB in the E plane. The superstrate technique provides again enhancement of 2.7 dB, reaching 16.5 at 24 GHz, as shown in Figure 11.

## 4. Antenna Performance Evaluation

This section compares and analyses the simulated and measured results of the SRR-GAA. Figure 12 shows the photographs of the fabricated prototype and compares the simulated and measured VSWR of the SRR-PGAA with the superstrate. There is reasonable agreement between these antennas. The observed impedance bandwidths of the PGAA reach 37.7% at the frequency range of 24 GHz from 20 GHz to 29.3 GHz. There is a slight bandwidth difference between the simulated and measured results. This problem is mainly caused by connector soldering defects, metal roughness, and PCB manufacturing errors, which can result in frequency shift and loss, especially at millimetre-wave frequencies [28,29,30,31].

For radiation pattern evaluation, the proposed antenna was used as the receiver, and a conventional millimetre-wave horn antenna served as the transmitter. The transmitter and receiver antennas were kept a wavelength or more apart. To obtain accurate gain estimates, both antennas must be driven by a uniform plane wave. This means that a region free of unwanted reflections and transmissions should be considered. To eliminate any unwanted interference, millimetre-wave absorbers were installed all around the measurement equipment. The gain transfer technique was used to assess the antenna radiation characteristics of the antenna, and the results are presented in Figure 12 [32]. There is a clear correlation between the simulated and measured trends. Due to manufacturing tolerance, the presence of the VNA, and the metal antenna mounts inside the chamber, the measured results show minor variations. The radiation pattern of the proposed PGAA with an added superstrate is shown in Figure 13, both in measured and simulated form. The comparison shows strong agreement and highlights an impressive reduction in SLL, which is less than −15.

Finally, Table 2 shows the performance of the developed SRR-GAA compared with state-of-the-art designs. According to the comparison, our design has the shortest size while achieving the widest bandwidth and the best gain.

## 5. Design of MIMO Radar

### 5.1. MIMO Antenna Configuration

In the proposed radar system, nine elements (six transmitters and three receivers) were arranged to form a virtual array of eighteen closely spaced elements, as shown in Figure 14a. The radar sensor should be carefully designed to avoid reflections from traffic signs or bridges above the road at a height of at least 5 m. Accordingly, the number of antenna elements was chosen to satisfy this condition in the elevation direction. A relatively wide beamwidth of about 60° in the azimuth direction is required to detect adjacent paths. To design the antenna coverage area, we assume a perspective antenna operating at a central wavelength λ; $x_{m}$ is the position of the *Tx* antennas, where m varies from 1 to 6; and Rx antennas are placed in position $y_{n}$, where n varies from 7 to 9. Suppose an object is positioned at a distance r from the automotive radar. The corresponding virtual MIMO array factor FMIMO is obtained by multiplying the Tx array factor $\left({AF}_{Tx}\right)$ by the Rx array factor $\left({AF}_{Rx}\right)$, as given in Equation (2) [5].
(2)$${AF}_{MIMO}(e)={AF}_{TX}(\overset{⃑}{e}){AF}_{RX}(\overset{⃑}{e})=\frac{1}{N}e^{i\frac{4\pi }{\lambda }r}\sum_{m,n}\left({exp}^{\left(\frac{x_{m}+y_{n}}{2}\right).\overset{⃑}{e}}\right)$$

According to Equation (2), the Tx array factor is advised to be null at every grating lobe of  Rx, and vice versa, to achieve a low SLL for a specific MIMO arrangement. Therefore, the spacing between the Tx elements is chosen to be $\lambda_{0}$ for Tx and 1.5$\lambda_{0}$ for Rx, as shown in Figure 14a. Figure 14b illustrates the MIMO configuration, including the packaging (superstrate). The array factor and 3D radiation pattern of the proposed configuration are depicted in Figure 14c,d. According to the proposed MIMO configuration, the SLL of the virtual array is less than −20 dB along the effective monitoring angle from 60° to 120°, where the AFTX nulls cancel the AFRX grating lobe. The coupling between the closest Tx–Rx antennas is examined and shown in Figure 15, illustrating excellent isolation of more than 20 dB.

### 5.2. Radar Image Reconstruction

With M transmitters and N receivers in the linear array radar system, we assume that they are all located on the *x*-axis with ($x_{m}$, 0) and ($x_{n}$, 0), where m = 1, 2, ..., M and n = 1, 2, ..., N, [5]. The complex representation of the transmitted stepped frequency continuous wave (SFCW) signal is given by
(3)$$S_{T}\left(t\right)=\sum_{q=1}^{Q}e^{-j2\pi \left(f_{0}+\left(q-1\right)∆f\right)t}rect(t/T-q)$$
where Δ*f* is the frequency step, *Q* is the number of frequencies, and *T* is the dwell time of each frequency. Assuming a target located at ($x_{0}$, $y_{0}$), for the mth transmitter, nth receiver, and qth frequency, $f_{q}=f_{0}+(q-1)∆f$, the echo signal mixed with the transmitted signal can be written as follows:(4)$$S_{m,n,q}=\alpha exp(-j2\pi f_{q}\tau_{mn})$$
where $\tau_{mn}=(\sqrt{{\left(x_{0}-x_{m}\right)}^{2}+{y_{0}}^{2}}+\sqrt{{\left(x_{0}-x_{n}\right)}^{2}+{y_{0}}^{2}}/c)$ is the time delay of the received signal, *α* is the reflection coefficient of the target, and c is the velocity of light. For distributed target, the received signal can be expressed as follows:(5)$$S_{m,n,q}=\int_{x_{0}}^{}\int_{y_{0}}^{}\alpha (x_{0},y_{0})e^{-j2\pi f_{q}\tau_{mn}}{dx}_{0}{dy}_{0}$$

We utilised the time-domain back-projection (TDBP) algorithm. Moreover, the point spread function (PSF), which is always used to demonstrate the resolution and SLL of the radar system, is defined as follows:(6)$$PSF≜\sum_{q=1}^{Q}\sum_{m=1}^{M}\sum_{n=1}^{N}e^{j2\pi f_{q}{(\tau }_{mn}^{*}-\tau_{mn})}$$

Furthermore, the design procedure of a linear uniform array can be summarised as follows: First, with the range resolution requirement $\delta_{r}$, the bandwidth of the SFCW B is determined using $\delta_{r}=c/2B$. Secondly, with the azimuth resolution requirement $\delta_{a}$ at the given range $R_{0}$, the array length is determined using $L=\lambda R_{0}\delta_{a}$, where $\lambda =c/f_{H}$ is the wavelength, and $f_{H}$ is the highest frequency. The proposed radar consists of six TX and three RX antennas. The SFCW radars transmit 2048 frequencies within a bandwidth of 500 MHz with a centre at 24 GHz (K-band). The radar has a maximum range ambiguity of 613 m and an azimuth angular resolution of 0.2 degrees.

Figure 16 shows the simulated radar image acquired with the parameters of the MIMO automotive radar system where six targets are located at (x, y) locations of (1, 2), (1.5, 2.5), (−2, 3), (−3, 4), (0, 5), and (1, 6) from the radar system, while the position location is in metre. The proposed system shows a very good resolution in both range and azimuth directions, verifying the applicability of the proposed configuration for MIMO radar systems. In order to guarantee the precision and credibility of our results in terms of azimuth resolution, we carried out an investigation using two targets positioned in close proximity at a distance of 10 m range, with a separation of 0.2 degrees. As depicted in Figure 17, we were able to accomplish this distinction by detecting a significant 3 dB difference in the signals returned from each target. This notable difference serves as a clear indication of the system’s ability to discriminate between closely positioned objects and validates the azimuth resolution achieved.

## 6. Conclusions

This paper presented a simple design of a PGAA with SRR radiating elements at 24 GHz for UWB automotive short-range radar sensors. Its impedance bandwidth and antenna gain were improved by using an array of SRR unit cells and a dielectric superstrate above it. The results show that the proposed PGAA outperforms the traditional GAA with straight-line radiating parts in terms of impedance and radiation performance. Impedance matching and radiation parameters of the proposed antenna were measured. The SRR-GAA shown here has an impedance bandwidth of 37.7%. At 24 GHz, a fan-shaped radiation pattern is produced with a peak gain of 16.5 dB. The proposed array antenna has been validated as a viable option for UWB automotive short-range radar sensors.

## Figures and Tables

**Figure 1 sensors-23-07070-f001:**
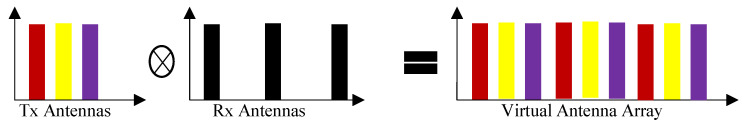
Convolution principle for MIMO radar (virtual array concept).

**Figure 2 sensors-23-07070-f002:**
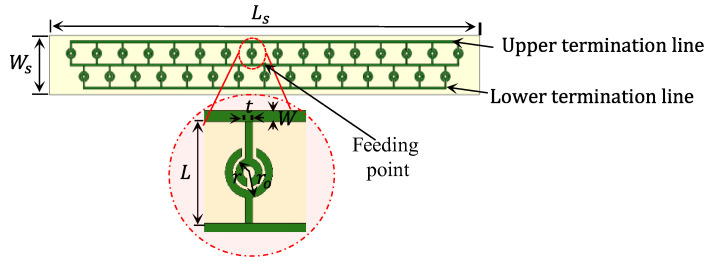
The proposed planar grid antenna array (PGAA).

**Figure 3 sensors-23-07070-f003:**
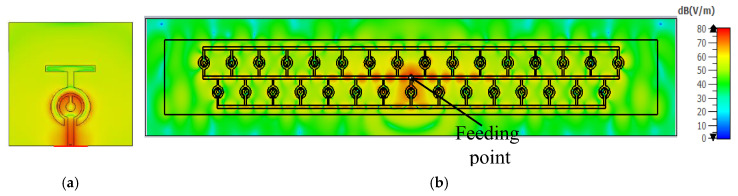
Electric field distribution at 24 GHz: (**a**) single element; (**b**) antenna array.

**Figure 4 sensors-23-07070-f004:**
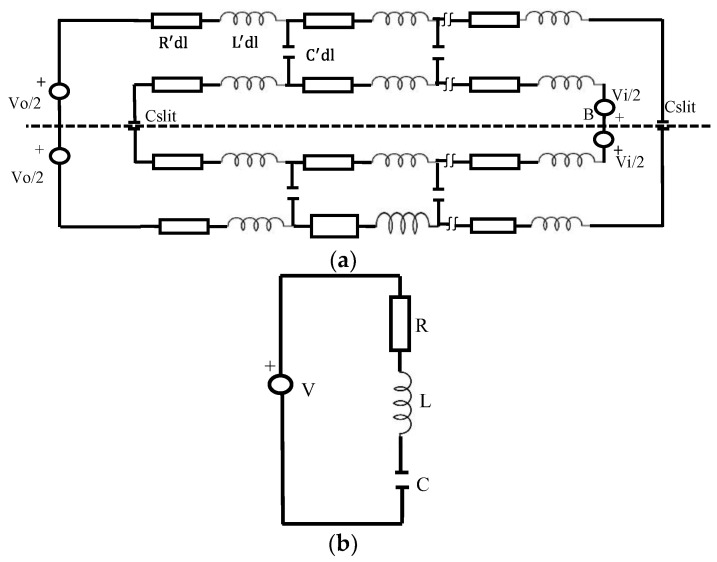
(**a**) The complete equivalent circuit of the SRR; (**b**) the simplified approximate equivalent circuit.

**Figure 5 sensors-23-07070-f005:**
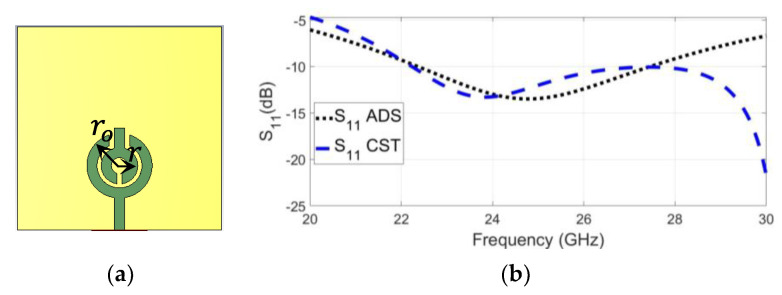
Single SRR element: (**a**) geometry; (**b**) reflection coefficients in CST and ADS versus frequency.

**Figure 6 sensors-23-07070-f006:**
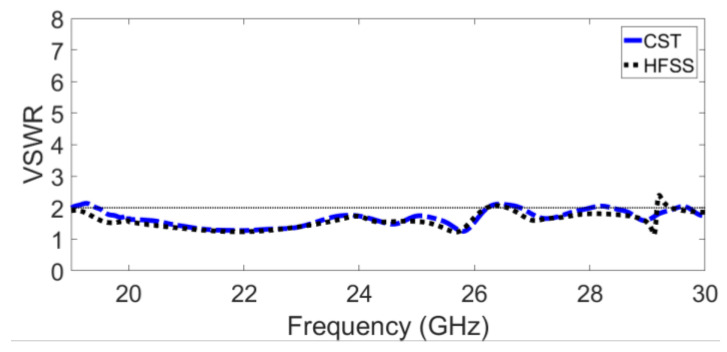
The simulated VSWR versus frequency.

**Figure 7 sensors-23-07070-f007:**
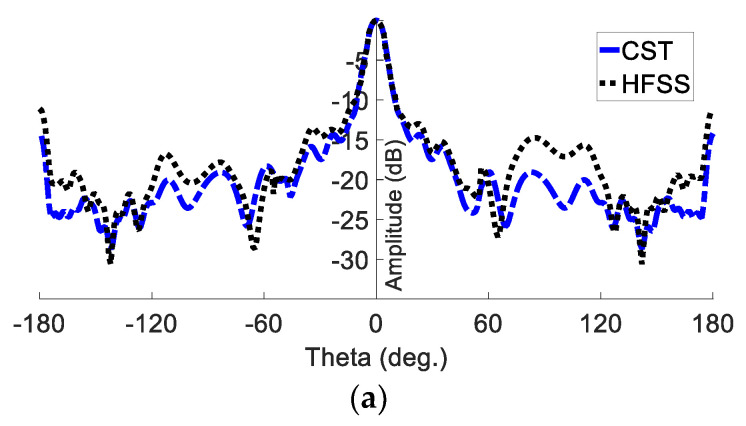
Simulated far-field pattern of the planner GAA at 24 GHz: (**a**) E plane; (**b**) H plane.

**Figure 8 sensors-23-07070-f008:**
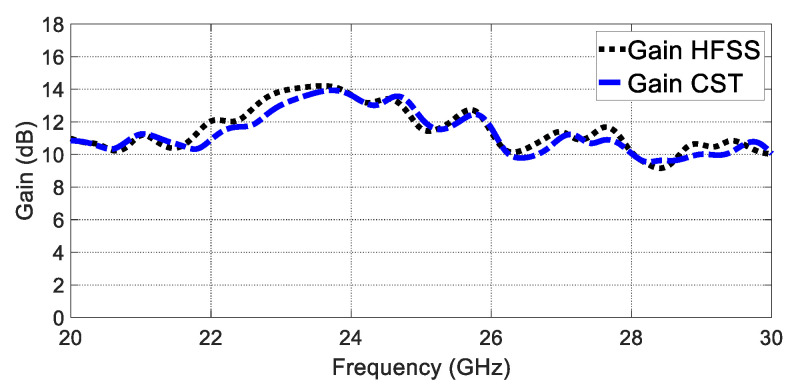
Gain versus frequency.

**Figure 9 sensors-23-07070-f009:**
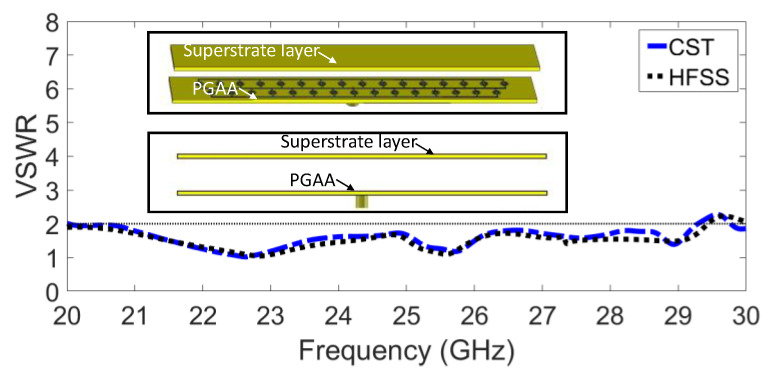
The simulated VSWR versus frequency using a superstrate dielectric layer.

**Figure 10 sensors-23-07070-f010:**
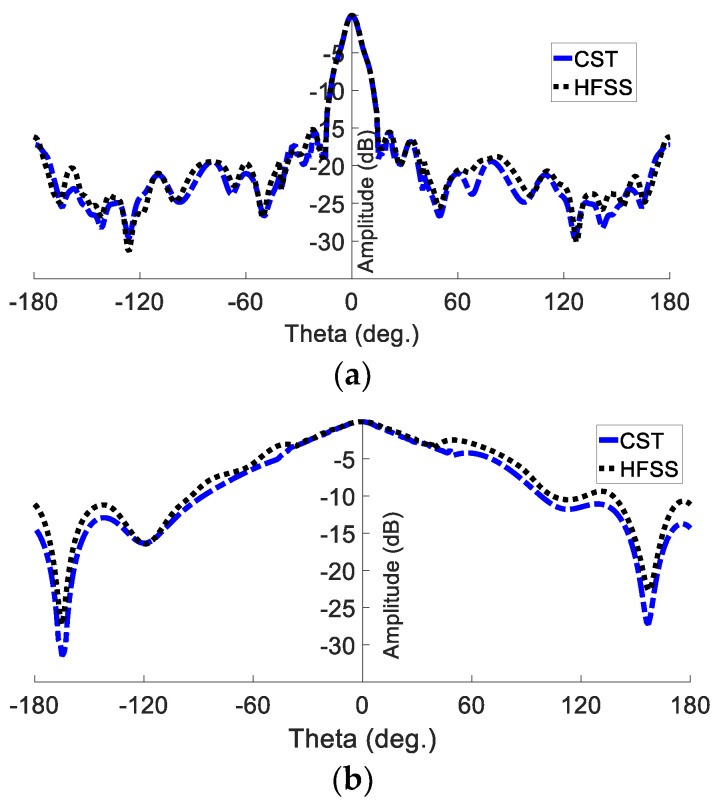
Simulated far-field pattern of PGAA with superstrate dielectric layer at 24 GHz: (**a**) E plane; (**b**) H plane.

**Figure 11 sensors-23-07070-f011:**
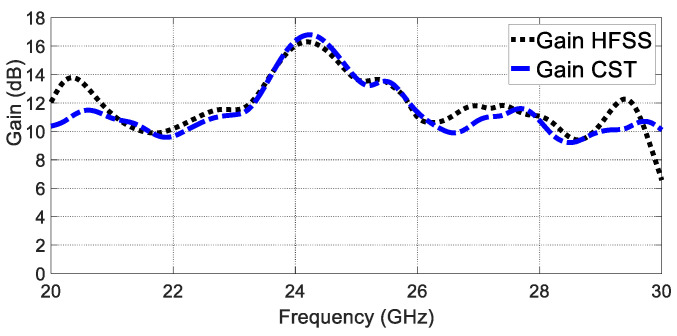
Gain with superstrate.

**Figure 12 sensors-23-07070-f012:**
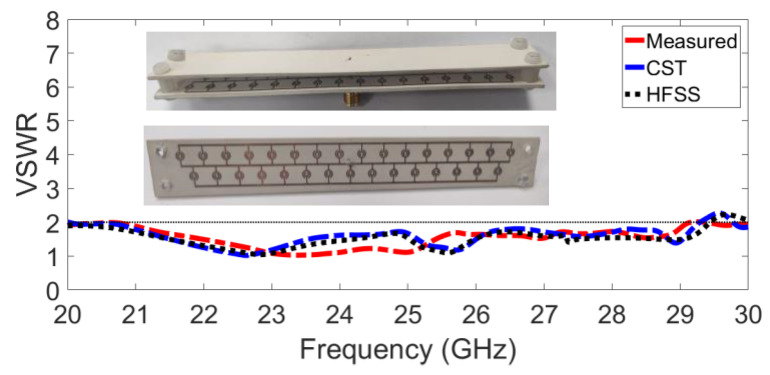
Measured and simulated VSWR for the proposed PGAA.

**Figure 13 sensors-23-07070-f013:**
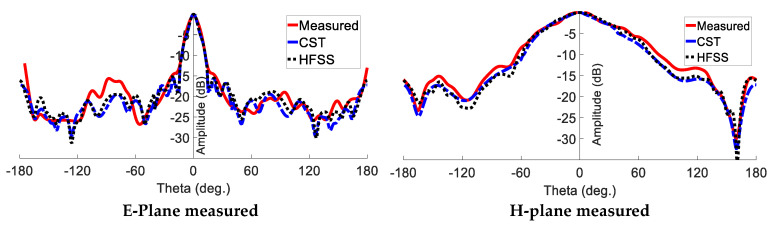
Measured and simulated radiation pattern at 24 GHz for the proposed PGAA.

**Figure 14 sensors-23-07070-f014:**
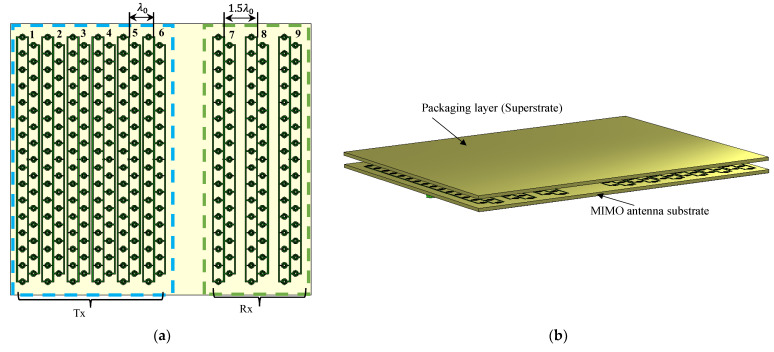
MIMO antenna configuration: (**a**) MIMO configuration; (**b**) MIMO antenna including packaging; (**c**) 3D radiation pattern; (**d**) MIMO radiation pattern array factor.

**Figure 15 sensors-23-07070-f015:**
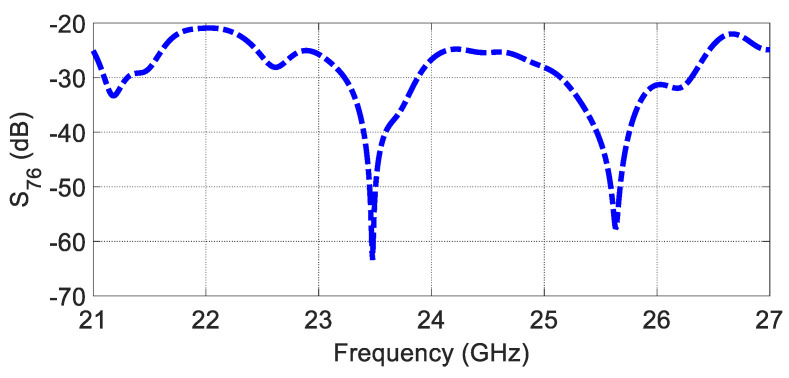
The mutual coupling between MIMO elements.

**Figure 16 sensors-23-07070-f016:**
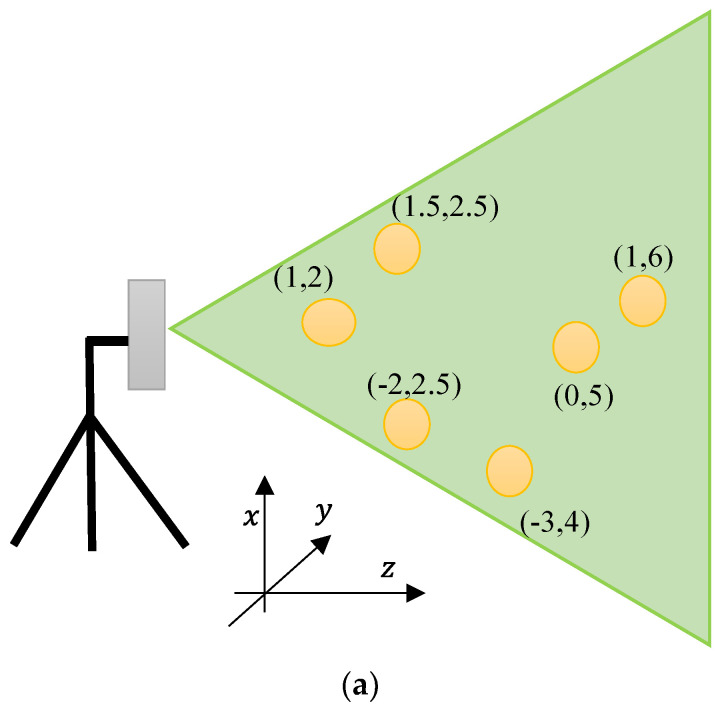
Automotive radar short-range image of six targets in different azimuth angles and range locations.

**Figure 17 sensors-23-07070-f017:**
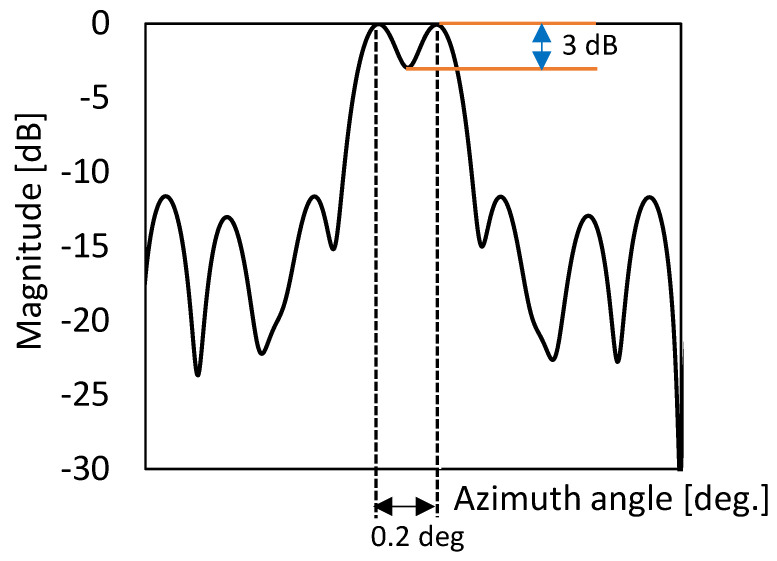
Automotive radar short-range image illustrating two targets located at the 10 m range but with a 0.2 deg difference in azimuth angle.

**Table 1 sensors-23-07070-t001:** Proposed antenna dimensions in mm.

L_s_	W_s_	L	W	t	r_o_	r
140	18	4.6	0.7	0.5	1.6	0.9

**Table 2 sensors-23-07070-t002:** Comparison with state-of-the-art designs.

Ref.	Frequency	Polarisation	Antenna Type	Size	Impedance B.W	Peak Gain (dBi)
[4]	24 GHz	Horizontal polarisation	mushroom-like Array	71.9 mm × 37 mm	16.9%	11.1
[17]	24 GHz	Linear	Grid Array	164 mm × 30 mm	25%	12.11
[33]	24 GHz	Linear	Grid Array	146 mm × 18 mm	20.88%	14.2
[18]	24 GHz	Linear	Grid array	146 mm × 18 mm	25%	11.59
Our work	24 GHz	Linear	Grid array	140 mm × 18 mm	37.7%	16.5

## Data Availability

The data supporting the findings of this study are available upon request from the corresponding author.

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
