# Peer review of "An Improved Performance Radar Sensor for K-Band Automotive Radars"

_sensors, 2023, doi:10.3390/s23167070_

Round 1

Reviewer 1 Report (Previous Reviewer 2)

Regarding the SRR model shown in Fig. 5 (a), the authors states to ignore the effect of the upper termination lines since numerous published papers have specifically addressed this issue by simulating the antenna with and without these lines [R1]. This is correct if the total antenna is considered. However, here the point is that a single element of total PGGA structure is modeled with an abruptly upper termination as in Fig. 5 (a), asserting that the effect is negligible as in total structure. At this point it would be useful to show for instance the field or current distribution to confirm the model goodness.

[R1] Zhang, L., Zhang, W. and Zhang, Y.P., 2011. Microstrip grid and comb array antennas. IEEE Transactions on Antennas and Propagation, 59(11), pp.4077-4084.

Author Response

Thank you for your instructive comment. We updated Fig.3, by adding the field distribution of the single element and add explanation for the field distribution highlighted in line 203.

Reviewer 2 Report (Previous Reviewer 1)

This paper presents a new radar sensor configuration of a planar grid antenna array (PGAA) for 12 automotive UWB radar applications. The simulation and measurement are carried out and excellent performance had been achieved. This paper can be accepted with current form.

Author Response

Thank you very much for your accepting our manuscripts.

Reviewer 3 Report (New Reviewer)

1- In Fig. 5, there is a large difference between CSt and ADS results. why?

2- How has the author chosen the number of antenna elements?

3- There are no measurement results related to the MIMO radar.

 4-Any reason behind this large bandwidth due to the proposed array?

Grammer may be improved in some places.

Author Response

Comment 1

- In Fig. 5, there is a large difference between CSt and ADS results. why?

Authors’ response:

The dissimilarity between CST and ADS results in Figure 5 could stem from divergent simulation algorithms and numerical methods used by each software. Additionally, variations in modelling assumptions and boundary conditions might contribute to the observed differences. Further investigation into these factors could help elucidate the specific reasons for the variance.

Comment 2

- How has the author chosen the number of antenna elements?

Authors’ response:

The determination of the antenna element quantity of the PGAA by the author stems from the necessary attributes of the radiation pattern, encompassing a 15-degree beamwidth in the E-plane and 60-degree beamwidth in the H-plane, along with a gain of 16.5 and a SLL below -15 dB. Regarding the MIMO configuration, the selection of TX and RX element counts is rooted in the demand for a synthetic aperture suitable for prospective automotive radar applications.

Comment 3

There are no measurement results related to the MIMO radar.

Authors’ response:

We acknowledge the comment regarding the absence of measurement results for the MIMO radar in our article. Due to equipment limitations in our labratory, we focused on rigorous simulations to evaluate and present the performance of the MIMO radar configuration. The simulated results provide valuable insights into its behavior and capabilities. We appreciate your understanding of this constraint and the effort we have put into the simulation-based analysis.

Comment 3

-Any reason behind this large bandwidth due to the proposed array?

Authors’ response:

The substantial bandwidth is a result of our careful choice of radiation elements. The selection of SRR radiating elements is based on their well-recognized attributes of broad impedance bandwidth and significant gain [30]. Comprising two conductive rings with a narrow gap, the SRR is imprinted on the identical dielectric substrate. Each ring incorporates a slit, arranged in a manner where the slits reside on opposing sides of the symmetry line. This conceptualizes the SRR as a pair of interconnected miniature loop antennas functioning across neighboring frequency bands.

We highlight this part line 134.

English grammar is also checked for he hole manuscript.

Reviewer 4 Report (New Reviewer)

The authors investigated the MIMO radar based on the planar array grid antenna composed of split ring resonators. They optimized the system for automotive application for working in K-band. The authors claimed that investigated system outperforms the conventional radars based on planar array grid antenna with the resonators in the form of patches, in terms of gain and impedance bandwidth.

The paper presents detailed results of the research: numerical simulations and measurements. The results obtained are valuable and interesting, although the subject of radars for automotive applications is widely explored and even the usage of split-ring resonators was already reported -  see the publication:

H. Cho, J. -H. Lee, J. -W. Yu and B. Ahn, "Series-Fed Coupled Split-Ring Resonator Array Antenna With Wide Fan-Beam and Low Sidelobe Level for Millimeter-Wave Automotive Radar," in IEEE Transactions on Vehicular Technology, vol. 72, no. 4, pp. 4805-4814, April 2023, doi: 10.1109/TVT.2022.3226294.  

The reviewer recommends the publication in MDPI Sensors after minor corrections:

1)      Please cite the paper by H. Cho (doi: 10.1109/TVT.2022.3226294).

2)      Introduce the following technical corrections:

-          line 45 – explain to which systems GAA are compared to be better,

-          please check if all abbreviations are introduces the their first appearance (e.g. UWB, GAA),

-         Sec.2. (page 3) please refer to the system presented in Sec.5.

-          Fig.7 – do not split part (a) and (b) of the figure between the pages,

-          Fig.10 -  both subfigures (a) and (b) should be of  the same sizes,

-          Line 344 (Eq.2) – please comment this equation properly. ‘e’ should be a unit vector pointing the direction at which the target is located; x_m and y_y denote (probably) the positions of  resonators in receiver and transmitter (please use the subscrips for m and n and use the same convention for denoting the positions of resonators as in Eqs. 3-6).

Author Response

Comment 1

Please cite the paper by H. Cho (doi: 10.1109/TVT.2022.3226294).

Authors’ response:

Done

Comment 2

Introduce the following technical corrections:

-    line 45 – explain to which systems GAA are compared to be better,

-    please check if all abbreviations are introduces the their first appearance (e.g. UWB, GAA),

-    Sec.2. (page 3) please refer to the system presented in Sec.5.

-     Fig.7 – do not split part (a) and (b) of the figure between the pages,

-     Fig.10 -  both subfigures (a) and (b) should be of  the same sizes,

-     Line 344 (Eq.2) – please comment this equation properly. ‘e’ should be a unit vector pointing the direction at which the target is located; x_m and y_y denote (probably) the positions of  resonators in receiver and transmitter (please use the subscrips for m and n and use the same convention for denoting the positions of resonators as in Eqs. 3-6).

Authors’ response:

All are Done

Round 2

Reviewer 3 Report (New Reviewer)

Satisfied with the author's response.

This manuscript is a resubmission of an earlier submission. The following is a list of the peer review reports and author responses from that submission.

Round 1

Reviewer 1 Report

This paper proposed, designed and measured a PGAA, the performance agrees well with the simulation and has some obvious merits. Some comments are as follows:

1, For Fig.2 and 3, please add some more details about the proposed model, such as its input and output ports.

2, For Fig.4, the authors said it was simulated by CST, but In Fig. 5, the results are from ADS?

3, Please give more explaining how to get the SRR unit from the circuit of Fig.4.

4, In Fig.9, Please label the dielectric layer in Fig.9.
